# Analytical Prediction of Residual Stress in the Machined Surface during Milling

**Caixu Yue [1],\*, Xiaole Hao [1], Xia Ji [2], Xianli Liu [1], Steven Y. Liang [3], Lihui Wang [4] and Fugang Yan [1]**

[1] Key Laboratory of Advanced Manufacturing and Intelligent Technology, Ministry of Education, Harbin University of Science and Technology, No. 52 Xuefu Road, Nangang 150080, China; 18348550143@163.com (X.H.); xlliu@hrbust.edu.cn (X.L.); yfg2988@163.com (F.Y.)

[2] School of Mechanical Engineering, Donghua University, No. 2999 North Renmin Road, Shanghai 201620, China; jixia0206@163.com

[3] Woodruff School of Mechanical Engineering, Georgia Institute of Technology, Atlanta, GA 30332, USA; steven.liang@me.gatech.edu

[4] Department of Production Engineering, KTH Royal Institute of Technology, 10044 Stockholm, Sweden; lihui.wang@iip.kth.se

\* Correspondence: yuecaixu@hrbust.edu.cn; Tel.: +86-1884-693-9745

**Abstract:** An analytical prediction model for residual stress during milling is established, which considers the thermal-mechanical coupling effect. Considering the effects of thermal-mechanical coupling, the residual stress distribution in the workpiece is determined by the stress loading history according to McDowell's hybrid algorithm. Based on the analysis of the geometric relationship of orthogonal cutting, the prediction model for milling force and residual stress in the machined surface is established. The research results can provide theoretical basis for stress control during milling.

**Keywords:** milling force; residual stress; analytical model; thermal-mechanical coupling; milling process

---

## 1. Introduction

The residual stress in the machined surface is a self-balanced internal stress. It is the internal stress remaining in the object after the elastoplastic object undergoes a large deformation and restores the external load, torque and thermal gradient to the initial state of the object. For components with weak rigidity, with the release of residual stress, the component will have obvious bending or torsional deformation, and the fatigue life of the component under cyclic loading will be reduced. The degree of effect of residual stress on the performance of components is generally judged by three indicators, that is, the residual stress property of the workpiece surface, the peak value of the residual stress and its corresponding depth on the subsurface. For the cutting process, the residual stress caused by plastic deformation, non-uniform heating and microstructure transformation are mainly studied.

The analytical method is efficient in calculation and can explain the physical mechanism clearly. Therefore, the research and exploration on the analytical algorithm of residual stress have become a difficult problem that many scholars insist to overcome. Su et al. [1] proposed an analytical model of residual stress during milling considering the corner radius under minimal lubrication conditions. The orthogonal model was applied to the more complex milling prediction by geometric transformation. Fergani et al. [2,3] established an analytical model for residual stress regeneration in milling and predicted the residual stress distribution in multi-pass milling and proposed an analytical model to predict the deformation of thin-walled parts caused by residual stress. Wang [4] established an analytical model of residual stress for flank milling. The model applied the radial return method to update the plastic stress components during the plastic loading process. Finally, the measurement

experiments for milling temperature and residual stress are performed to validate the analytical model. It is found that the plowing effect of the cutting edge on the workpiece is the major source of the residual stress. Wan et al. [5] considered the 3D instantaneous contact status between the tool and the workpiece to predict the residual stress during milling. A modified analytical model for predicting the milling temperature field was presented. Huang and Yang [6] proposed a criterion for determining the initial stress state of the workpiece based on the calibration of the residual stress and achieved accurate prediction of the residual stress distribution of workpieces of different materials. Huang [7] proposed an influence coefficient method for quickly calculating the residual stress distribution based on the eigenstrain theory. The analytical model established by this method can predict the periodic residual stress distribution under dynamic cutting conditions. The analytical method has high-efficiency prediction and can directly reflect the physical mechanism of cutting, which can effectively promote the development of finite element methods for residual stress. In addition, it can guide the direction of experimental research and establish the relationship between residual stress and other data that can be easily obtained by experiments.

The residual stress is greatly affected by the cutting forces and temperature which they influence each other. However, the milling process has the characteristic of discontinuity, the force and heat acting in the surface of the workpiece are two complex amounts that change periodically, it is difficult to consider the coupling of milling force and milling heat. In this study, iterative algorithm of thermal-mechanical coupling in orthogonal cutting is introduced, and the prediction of residual stress in milling process is simplified to takes both mechanical effect and thermal effect into consideration. The research objective is to accurately predict the peak value of residual stress on the subsurface of Ti6Al4V, and realize the prediction of residual stress using MATLAB® based program (The software version is R2016a, Natick, MA, USA). The main innovations of this paper are as follows:

(1) Considering the thermal-mechanical coupling, a milling force model is established. Based on the thermo-mechanical coupling algorithm, the relative motion relationship between the tool and the workpiece during milling is analyzed, and a milling force model considering the thermal-mechanical coupling effect is established.
(2) Based on the milling force model, the residual stress prediction model is established. Considering the surface waviness of the workpiece after milling, an approximate assumption is proposed to planarize the machined surface to adapt to the residual stress analysis algorithm.

## 2. Research on Iterative Algorithm of Thermal-Mechanical Coupling in Orthogonal Cutting

Ji Xia [8] proposed an iterative algorithm of thermal-mechanical coupling in orthogonal cutting. The cutting force predicted by the Oxley's cutting force model is used as the input of the cutting temperature prediction model to obtain the cutting temperature, and then put it into the Oxley's cutting force model to obtain a new cutting force. After several iterations, when the output cutting force converges to a certain value, at this time, it is considered that the cutting force and the cutting temperature have reached a balance, which is the process of thermal-mechanical coupling.

The Oxley's cutting force model considers that the strain rate on the shear zone is almost constant. This view was confirmed by Stevenson and Oxley [9]. Furthermore, it can be considered that the flow velocity $V_S$ on the shear zone is constant. In addition, it is assumed that the flow stress on the shear zone is uniformly distributed at a certain instant. So the cutting force can be calculated based on the uniformly distributed stress on the shear zone, and the cutting mechanism is analyzed based on the geometric relationship. The assumptions of the model are as follows:

(1) There is no tool wear and cutting vibration, and no built-up edge is generated.
(2) Plastic deformation in the cutting layer only occurs in a plane perpendicular to the cutting edge.
(3) The primary and secondary deformation zones are assumed to be narrow area.
(4) The temperature and strain on the shear plane are uniformly distributed.

The geometric relationship of force and material flow speed in three different directions (along the cutting direction, along the shear plane AB and along the tool–chip interface) is shown in Figure 1. The resultant cutting force $P_{total}$ can be expressed by the flow stress $\sigma_{AB}$ on the shear plane AB by the geometric relationship as:

$$P_{total} = \frac{\sigma_{AB} h_c w}{\sin \Phi \cos \theta} \tag{1}$$

where $h_c$ is the cutting thickness, $w$ is the width of the cutting edge participating in cutting, $\Phi$ is the shear angle of the primary shear zone, and $\theta$ is the angle between the force $P_{total}$ and the shear plane AB. The friction $F$ and normal force $N$ at the rake face of the tool, the force $F_c$ in the cutting direction and the force $F_t$ in the feed direction can be expressed as:

$$\begin{cases} F = P_{total} \sin \lambda \\ N = P_{total} \cos \lambda \\ F_c = P_{total} \cos(\lambda - \alpha) \\ F_t = P_{total} \sin(\lambda - \alpha) \end{cases} \tag{2}$$

where $\alpha$ is the rake angle of the tool and $\lambda$ is the friction angle of the rake face.

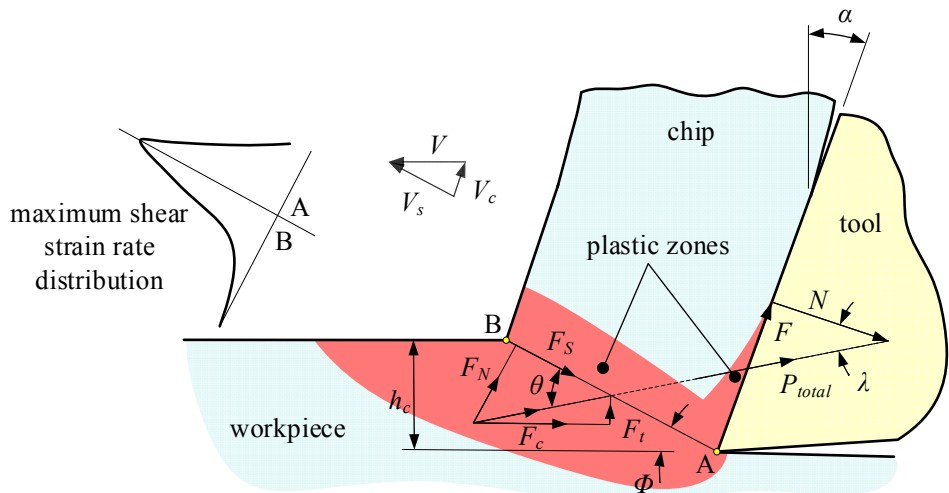

**Figure 1.** Schematic diagram of the geometric relationship of orthogonal cutting (adapted from [9], with permission from The Royal Society, 2020).

Shear flow stress $\sigma_{AB}$ can be obtained from various constitutive models, such as the Johnson–Cook constitutive model, Zerilli–Armstrong constitutive model, Bammann–Chiesa–Johnson constitutive model, etc. Hyperbolic tangent (TANH) constitutive model is applied to predict the shear flow stress at shear zone. The constitutive model is proposed by Calamaz [10]. It considers the strain softening effect of Ti6Al4V on the basis of J-C constitutive model. According to the study of Ducobu [11], the constitutive parameters are shown in the Table 1.

**Table 1.** Parameters of the TANH constitutive model [11].

| *A* (MPa) | *B* (MPa) | *n* | *C* | *m* | $\dot{\varepsilon}_0$ | *a* | *b* | *c* | *d* | $T_m$ (°C) |
|---|---|---|---|---|---|---|---|---|---|---|
| 968 | 380 | 0.421 | 0.0197 | 0.577 | 0.1 | 1.6 | 0.4 | 6 | 1 | 1650 |

Since many parameters in the Oxley's cutting force model need to be determined by experiments, Ji Xia [8] proposed an iterative algorithm to capture the parameters that need to be experimentally defined in the Oxley's cutting force model. This algorithm makes the input parameters of the cutting

force model include only material parameters, process parameters and tool geometry parameters, which improves the predictive efficiency.

The model developed by Huang and Liang [12] is used in the present study to predict the cutting temperature. This model considers the combined effect of the heat source in primary and secondary deformation zones, the non-uniform heat distribution ratio along the chip contact surface to obtain the temperature along the tool–chip interface. In the present study, the temperature generated by the friction in the tertiary deformation zone is considered. The heat source in the tertiary deformation zone is calculated based on the model proposed by Su [1]. It is assumed that the heat distribution ratio in the tertiary deformation zone is constant. The temperature of chips and tools is calculated by the following equation.

$$\Delta T_{c-s}\left(X^{ch}, Z^{ch}\right) = \frac{q_{shear}}{2\pi K_c} \int_{l_i=0}^{L_{AB}} e^{\frac{-(X^{ch}-x_i)V_c}{2a_c}} \left\{ K_0\left(\frac{R_i V_c}{2a_c}\right) + \frac{1}{2}K_0\left(\frac{R'_i V_c}{2a_c}\right) + \frac{1}{2}K_0\left(\frac{R''_i V_c}{2a_c}\right) \right\} dl_i \tag{3}$$

$$\Delta T_{c-f}\left(X^{ch}, Z^{ch}\right) = \frac{q_f}{\pi K_c} \int_{l_i=0}^{h} B(x) e^{\frac{-(X^{ch}-x)V_c}{2a_c}} \left\{ K_0\left(\frac{R_i V_c}{2a_c}\right) + K_0\left(\frac{R'_i V_c}{2a_c}\right) + K_0\left(\frac{R''_i V_c}{2a_c}\right) \right\} dx \tag{4}$$

$$\Delta T_{t-f}\left(X^t, Y^t, Z^t\right) = \frac{q_f}{2\pi K_t} \int_{0}^{h} (1 - B(x)) \int_{-w/2}^{w/2} \left(\frac{1}{R_i} + \frac{1}{R'_i}\right) dy dx \tag{5}$$

$$\Delta T_{t-rub}\left(X^t, Y^t, Z^t\right) = \frac{q_{rub}}{2\pi K_t} \int_{0}^{CA} (1 - \gamma) \int_{-w/2}^{w/2} \left(\frac{1}{R_i} + \frac{1}{R'_i}\right) dy dz \tag{6}$$

where $R_i$, $R'_I$, $R''_i$ are the distances between any point on the workpiece and a certain differential unit on the heat source. $B(x)$ is the percentage of heat generated at the tool–chip interface entering the chip. $1 - \gamma$ is the percentage of heat generated by the heat source $q_{rub}$ entering the tool.

Assuming that the temperature at the side of the tool and the temperature at the side of the chip are consistent on the adiabatic boundary, the following relationship exists at the tool–chip interface.

$$\Delta T_{c-s} + \Delta T_{c-f} + T_0 = \Delta T_{t-f} + \Delta T_{t-rub} + T_0 \tag{7}$$

$B(x)$ is calculated by Equation (7) and the temperatures in the primary shear zone and the temperature at the tool–chip interface are obtained by Equations (3) and (4), respectively.

## 3. Milling Force Prediction Based on Analytical Method

When the milling force model is established, the cutting-edges of the tool are discretized along the axis of the tool, and the cutting process of each discrete cutting edge is regarded as an oblique cutting to calculate separately, as shown in Figure 2a,b. Milling force is mainly divided into two parts: chip forming force and ploughing force. The chip forming force mainly stems from the normal pressure $N$ and the friction force $F$ exerted by the chip on the tool. The ploughing force mainly stems from the force $P_{cut}$ in the cutting direction and the force $P_{thrust}$ perpendicular to the cutting direction caused by the relative movement between the tool tip and the workpiece, and the detailed calculations refer to the study of Su [1]. The cutting thickness and direction of the cutting force are different at different positions on the cutting-edge during milling. Considering the influence of chip side flow on force and temperature is challenging, so it is considered that the chip side flow angle is zero during milling which makes it easier to introduce the iterative algorithm of thermal-mechanical coupling in orthogonal cutting. Figure 2b illustrates the geometric relationship of the oblique cutting process without considering the phenomenon of chips side-flow. Obviously, the angle between the frictional

force $F$ and the radial force $F_r$ is the rake angle $\alpha$. Make a section that is perpendicular to the cutting edge, as shown in Figure 2c, the rotation matrix $M_1$ can be obtained as:

$$M_1 = \begin{bmatrix} \cos \alpha & -\sin \alpha & 0 \\ \sin \alpha & \cos \alpha & 0 \\ 0 & 0 & 1 \end{bmatrix} \tag{8}$$

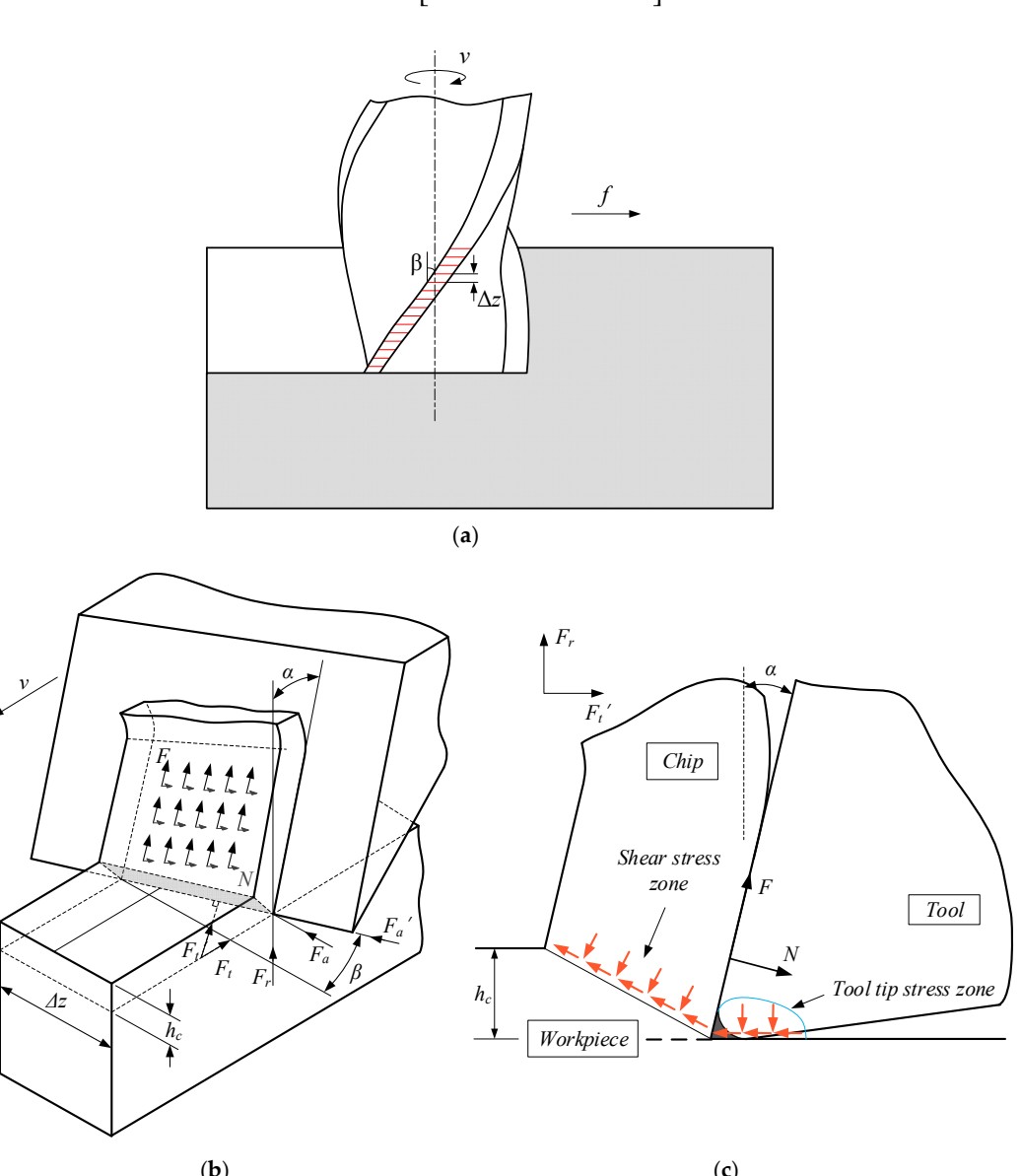

**Figure 2.** Schematic diagrams of the geometric relationship of the oblique cutting process. (**a**) Axial slicing of helical end mill; (**b**) Schematic diagrams of the oblique cutting process; (**c**) Schematic diagrams of section along the cutting edge.

Then the radial force $F_r$ and the circumferential force $F_t'$ perpendicular to the cutting edge are expressed as:

$$\begin{bmatrix} F_r \\ F'_t \\ 1 \end{bmatrix} = M_1 \begin{bmatrix} F \\ N \\ 1 \end{bmatrix} + \begin{bmatrix} P_{thrust} \\ P_{cut} \\ 1 \end{bmatrix} \tag{9}$$

As shown in Figure 2b, the rotation matrix $M_2$ considering the inclination angle $\beta$ can be obtained as:

$$M_2 = \begin{bmatrix} 1 & 0 & 0 \\ 0 & \cos\beta & 0 \\ 0 & \sin\beta & 0 \end{bmatrix} \tag{10}$$

Then the radial force $F_r$, the circumferential force $F_t$, and the axial force $F_a$ are expressed as:

$$\begin{bmatrix} F_r \\ F_t \\ F_a \end{bmatrix} = M_2 \begin{bmatrix} F_r \\ F'_t \\ 1 \end{bmatrix} \tag{11}$$

Figure 3 shows the trajectory of a point on the cutting edge with an axial height $z$ during milling. The rotation matrix $M_3$ considering the rotation angle of the tool can be obtained as:

$$M_3 = \begin{bmatrix} \sin(\psi_2 - \omega t + z\tan\beta/R) & \cos(\psi_2 - \omega t + z\tan\beta/R) & 0 \\ -\cos(\psi_2 - \omega t + z\tan\beta/R) & \sin(\psi_2 - \omega t + z\tan\beta/R) & 0 \\ 0 & 0 & 1 \end{bmatrix} \tag{12}$$

where $\omega$ is the tool speed, $t$ is the cutting time, $z$ is the axial height of the tool, $\beta$ is the tool helix angle, $R$ is the tool radius, $\psi_2$ can be expressed as:

$$\psi_2 = \frac{\pi}{2} - \arcsin\left(\frac{R - a_e}{R}\right) \tag{13}$$

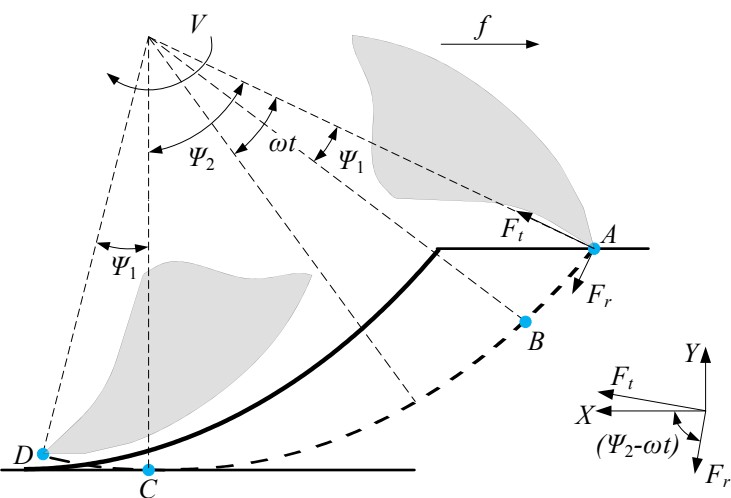

**Figure 3.** Geometric diagram of milling process.

Then the cutting force $dF_X$ along the feed direction, the cutting force $dF_Y$ in the direction perpendicular to the workpiece surface, and the tool axial cutting force $dF_Z$ on the micro cutting edge are expressed as:

$$\begin{bmatrix} dF_X(z,t) \\ dF_Y(z,t) \\ dF_Z(z,t) \end{bmatrix} = M_3 \begin{bmatrix} F_r \\ F_t \\ F_a \end{bmatrix} = M_3 M_2 \left( M_1 \begin{bmatrix} F \\ N \\ 1 \end{bmatrix} + \begin{bmatrix} P_{thrust} \\ P_{cut} \\ 1 \end{bmatrix} \right) \tag{14}$$

Assuming that the milling process is down milling, the total cutting time $T_c$ is divided into three stages, as shown in Figure 3. The first stage is the cut-in stage. A part of the cutting edge is in contact with the workpiece, which occurs during the point on the end of the tool moves from A to B. The second stage is the stable cutting stage. The cutting edge is completely in contact with the workpiece, which occurs during the point on the end of the tool moves from B to C. The third stage is the cut-out stage. A part of the cutting edge is in contact with the workpiece, which occurs during the point on the end of the tool moves from C to D.

The total milling force is calculated based on three cutting stages, and its expression is:

$$F_j^{(n)} = \sum_{i=1}^{n} dF_j(i\Delta z, t) \quad j = x, y, z \tag{15}$$

where $n$ is the number of micro cutting edges involved in the cutting process.

The cutting thickness $h_c$ during milling can be considered as a function of the independent variable $t$. Figure 4 is a geometric schematic of the cutting thickness, where points $O_1$ and $O_2$ are the positions of the tool axis at time $t_1$ and $t_2$ respectively, and point O is the position of the tool axis when the cutting edge cuts into the workpiece. A reference coordinate system XOY is established with the point O as the origin. The cutting thickness $h_c$ is defined as the length of the workpiece passing between the point on the cutting edge and the axis of the tool, and its expression is:

$$h_c = \sqrt{(y_2 - y_1)^2 + (x_2 - x_1)^2} \tag{16}$$

where $(x_2, y_2)$ is the coordinate of the cutting edge, $(x_1, y_1)$ is the coordinate of a point on the workpiece surface.

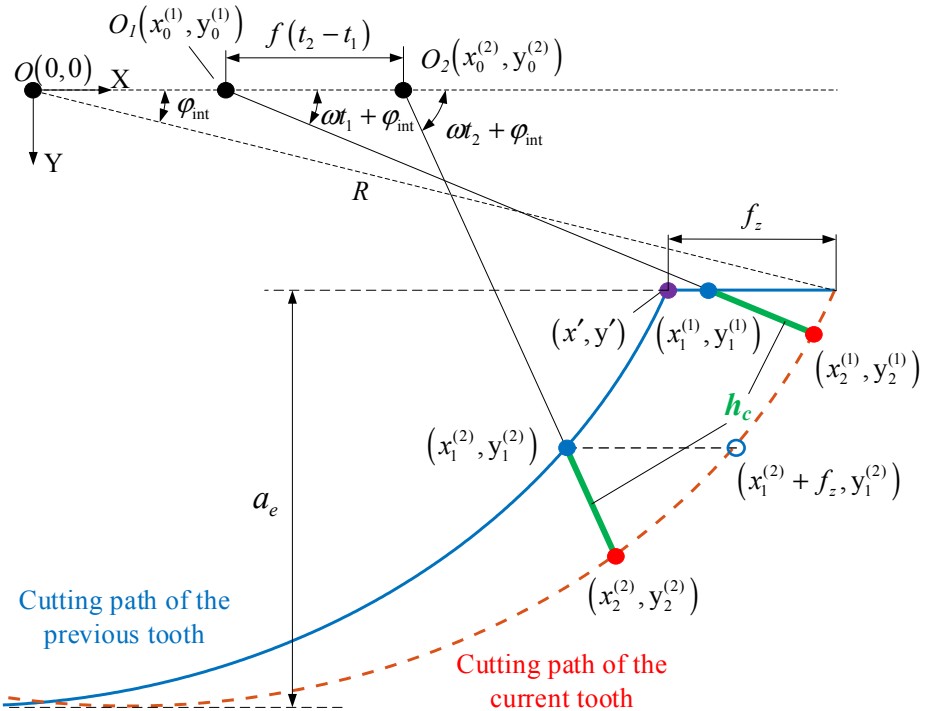

**Figure 4.** Schematic diagram of the geometric relationship of cutting thickness.

As shown in Figure 4, when the coordinate of a point on the cutting edge is $\left(x_2^{(1)}, y_2^{(1)}\right)$, the cutting thickness $h_c$ is the distance between the cutting edge and the surface which is to be machined, and when the coordinate of a point on the cutting edge is $\left(x_2^{(2)}, y_2^{(2)}\right)$, the cutting thickness $h_c$ is the distance between the cutting edge and the machining surface. Coordinate point $(x', y')$ is used as the judgment standard, which can be expressed by the cutting amount and the geometric angle of the tool as:

$$\begin{cases} x' = f \cdot t + R \cos \varphi_{\text{int}} - f_z \\ y' = R - a_e \end{cases} \tag{17}$$

where $f_z$ is the feed per tooth, $\varphi_{\text{int}}$ is the cut-in angle, which is obtained from the geometric relationship and its expression is: $\varphi_{\text{int}} = \arcsin((R - a_e)/R)$.

Assuming that the cutting process is ideal without vibration and tool wear, it can be considered that the distance in the horizontal direction between the cutting path of the previous tooth and the cutting path of the current tooth is $f_z$. In the reference coordinate system XOY, the current path can be expressed as:

$$\begin{cases} x_2 = f \cdot t_2 + R\cos(\omega t_2 + \varphi_{\text{int}}) \\ y_2 = R\sin(\omega t_2 + \varphi_{\text{int}}) \end{cases} \tag{18}$$

As shown in Figure 4, A and B are on the same straight line. If the slope of the straight line is $k$, the equation can be obtained:

$$\frac{R\sin(\omega t_1 + \varphi_{\text{int}}) - 0}{[f \cdot t_1 + R\cos(\omega t_1 + \varphi_{\text{int}}) - f_z] - f \cdot t_2} = \frac{R\sin(\omega t_2 + \varphi_{\text{int}}) - 0}{[f \cdot t_2 + R\cos(\omega t_2 + \varphi_{\text{int}})] - f \cdot t_2} = k \tag{19}$$

where $t_1$ is the time for the cutting edge to move to $\left(x_1^{(2)}, y_1^{(2)}\right)$ on the cutting path of the previous cutter tooth.

Calculate $t_1$ by Equation (19) and substitute it into Equation (20) to calculate $\left(x_1^{(2)}, y_1^{(2)}\right)$.

$$\begin{cases} x_1^{(2)} = f \cdot t_1 + R\cos(\omega t_1 + \varphi_{\text{int}}) - f_z \\ y_1^{(2)} = R\sin(\omega t_1 + \varphi_{\text{int}}) \end{cases} \tag{20}$$

Finally, $\left(x_1^{(2)}, y_1^{(2)}\right)$ and $\left(x_2^{(2)}, y_2^{(2)}\right)$ are substituted into Equation (16) to calculate the cutting thickness $h_c$. When the cutting thickness $h_c$ is the distance between the cutting edge and the surface which is to be machined, since $y_1^{(1)}$ is known, $x_1^{(1)}$ can be obtained by Equation (21).

$$\frac{(R - a_e) - 0}{x_1^{(1)}} = \frac{R\sin(\omega t_2 + \varphi_{\text{int}}) - 0}{[f \cdot t_2 + R\cos(\omega t_2 + \varphi_{\text{int}})] - f \cdot t_2} = k \tag{21}$$

The calculation process of the milling force is shown in Figure 5.

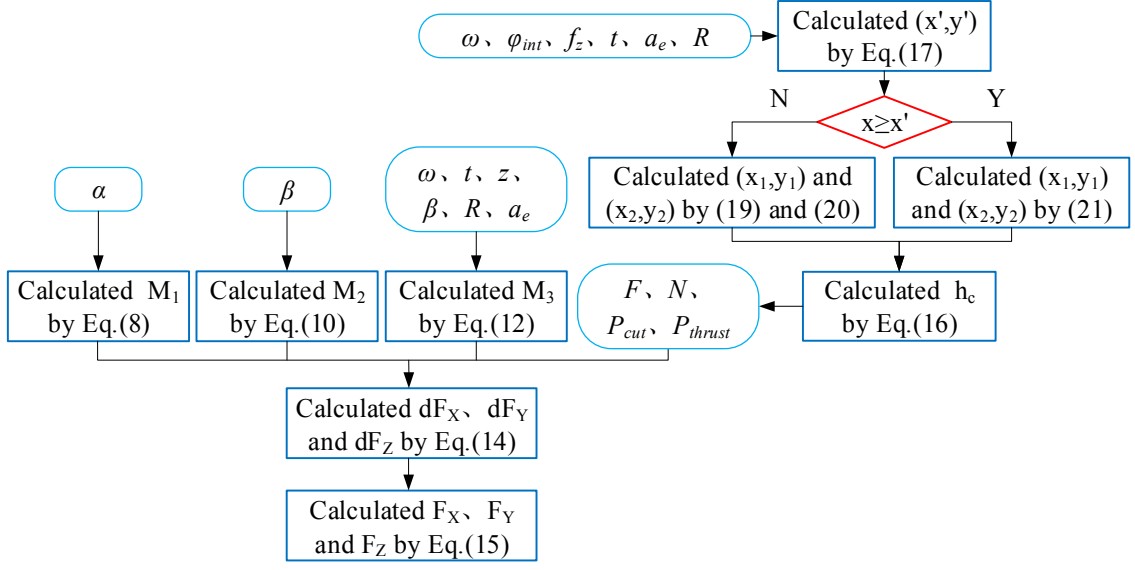

**Figure 5.** Flow chart of milling force calculation.

## 4. Prediction of Residual Stress Considering Thermal-Mechanical Coupling Effect

Su and Liang [1] have developed an analytical modeling of residual stress for dry machining which considers both mechanical effect and thermal effect. In this paper, the iterative algorithm of thermal-mechanical coupling in orthogonal cutting is introduced, and the geometry of the workpiece surface is simplified to calculate the residual stress more effectively, so as to realize the rapid prediction of the peak value of the residual stress.

### 4.1. Mechanical Stress Source and its Stress Distribution Model

When calculating the residual stress, the stress loading history inside the workpiece needs to be known and it can be calculated by Hertzian rolling contact theory. Compared with the primary and tertiary deformation zones, the effect of the secondary deformation zone on the stress distribution of the workpiece is indirect and can be ignored. Therefore, the stress due to mechanical load during cutting is mainly two aspects: one is the stress caused by the shear deformation of the workpiece, the other is the stress caused by the friction and extrusion of the tool tip and the surface of the machined workpiece, as shown in Figure 2c.

The contact between the tool and the workpiece is regarded as Hertzian rolling/sliding contact. At the point $(x, z)$ in the workpiece, the stress component generated by the mechanical load can be obtained by integrating the Boussinesq solution in the loaded area $(-a < x < a)$:

$$\begin{cases} \sigma_x = -\frac{2z}{\pi} \int\limits_{-a}^{a} \frac{p(s)(x-s)^2}{\left[(x-s)^2+z^2\right]^2} ds - \frac{2}{\pi} \int\limits_{-a}^{a} \frac{q(s)(x-s)^3}{\left[(x-s)^2+z^2\right]^2} ds \\ \sigma_z = -\frac{2z^3}{\pi} \int\limits_{-a}^{a} \frac{p(s)}{\left[(x-s)^2+z^2\right]^2} ds - \frac{2z^2}{\pi} \int\limits_{-a}^{a} \frac{q(s)(x-s)}{\left[(x-s)^2+z^2\right]^2} ds \\ \sigma_{xz} = -\frac{2z^2}{\pi} \int\limits_{-a}^{a} \frac{p(s)(x-s)}{\left[(x-s)^2+z^2\right]^2} ds - \frac{2z}{\pi} \int\limits_{-a}^{a} \frac{q(s)(x-s)^2}{\left[(x-s)^2+z^2\right]^2} ds \end{cases} \tag{22}$$

where $2a$ is the length of the loading range of the mechanical load, $p(s)$ and $q(s)$ are the normal stress and the tangential stress on the loading area. $\sigma_x$ is the stress along the feed direction and $\sigma_y$ is the stress along the direction of tool axis.

The normal stress, tangential stress, and loading range on the shear plane are expressed as:

$$p(s) = \frac{F_c \sin \Phi + F_t \cos \Phi}{L_{AB} w} \quad q(s) = \sigma_{AB} \quad a = \frac{1}{2} L_{AB} \tag{23}$$

The normal stress, tangential stress, and loading range on the friction area between the tool tip and the workpiece are expressed as:

$$p(s) = \frac{2P_{thrust}}{\pi(wa)} \quad q(s) = \mu\left(\frac{P_{cut}}{w \cdot CA}\right) \quad a = \frac{1}{2}CA \tag{24}$$

### 4.2. Thermal Stress Source and its Stress Distribution Model

The stress due to thermal load during cutting is mainly two aspects: one is the shear heat source generated on the shear plan, the other is the friction heat source generated by the friction between the tool tip and the surface of the processed workpiece. Thermal stress caused by temperature changes is expressed as:

$$\begin{cases} \sigma_x^{therm}(x,z) = -\frac{\alpha_T E}{1-2v} \int_0^\infty \int_{-\infty}^\infty \left(G_{xh}\frac{\partial T}{\partial x}(x,z) + G_{xv}\frac{\partial T}{\partial x}(x,z)\right) dx dz + \frac{2z}{\pi} \int_0^\infty \frac{p(t)(t-x)^2}{\left((t-x)^2+z^2\right)} dt - \frac{\alpha_T E T(x,z)}{1-2v} \\ \sigma_z^{therm}(x,z) = -\frac{\alpha_T E}{1-2v} \int_0^\infty \int_{-\infty}^\infty \left(G_{zh}\frac{\partial T}{\partial x}(x,z) + G_{zv}\frac{\partial T}{\partial x}(x,z)\right) dx dz + \frac{2z^3}{\pi} \int_0^\infty \frac{p(t)}{\left((t-x)^2+z^2\right)} dt - \frac{\alpha_T E T(x,z)}{1-2v} \\ \sigma_{xz}^{therm}(x,z) = -\frac{\alpha_T E}{1-2v} \int_0^\infty \int_{-\infty}^\infty \left(G_{xzh}\frac{\partial T}{\partial x}(x,z) + G_{xzv}\frac{\partial T}{\partial x}(x,z)\right) dx dz + \frac{2z^2}{\pi} \int_0^\infty \frac{p(t)(t-x)}{\left((t-x)^2+z^2\right)} dt \end{cases} \tag{25}$$

where $E$ is the elastic modulus of the workpiece material, $v$ is the Poisson's ratio of the workpiece material, $\alpha_T$ is the thermal diffusion coefficient of the workpiece material, $G_{xh}$, $G_{xv}$, $G_{zh}$, $G_{zv}$, $G_{xzh}$ and $G_{xzv}$ represent the Green functions of the plane strain, and the detailed calculations refer to the study of Saif [13]. $p(t)$ is expressed as:

$$p(t) = \frac{\alpha ET(x, z = 0)}{1 - 2v} \tag{26}$$

The stress at any point on the workpiece due to mechanical and thermal loads is expressed as:

$$[\sigma_{total}] = \left[\sigma_{total}^{mech}\right] + \left[\sigma_{total}^{thermal}\right] \tag{27}$$

Since the milling is discontinuous cutting, the cutting thickness and the cutting direction of the cutting-edge change at all times. Its stress distribution is completely different from that of turning. As shown in Figure 6, plastic deformation occurs at the contact area between the tool and the workpiece during milling. Obviously, the plastic deformation between the A and B points has the largest effect on the residual stress in the machined surface. At the same time, the surface between points A and B is simplified to a plane for calculation. This simplification is reasonable because the surface roughness $Rz$ is much smaller than the feed rate per tooth $f_z$, Furthermore, the calculation of the surface stress distribution of the workpiece is simplified to a plane calculation between A and B points, which reduces the workload of the residual stress calculation and improves the predictive efficiency. Taking the experimental parameters of case 1 in Table 2 as an example, Figure 7 shows the stress contours in the workpiece due to the combined loads.

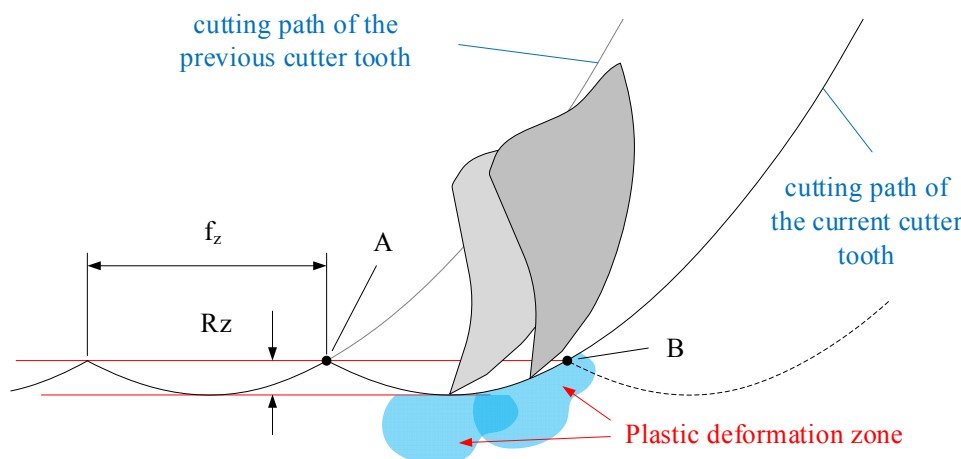

**Figure 6.** Schematic diagram of plastic deformation zone distribution in the surface and subsurface of the workpiece during milling.

**Table 2.** The experimental scheme.

| Case | Spindle Speed (r/min) | Cutting Width (mm) | Cutting Depth (mm) | Feed Rate (mm/min) |
|------|------|------|------|------|
| 1 | 750 | 0.1 | 8 | 100 |
| 2 | 600 | 0.1 | 8 | 100 |
| 3 | 750 | 0.12 | 8 | 100 |
| 4 | 750 | 0.1 | 8 | 120 |

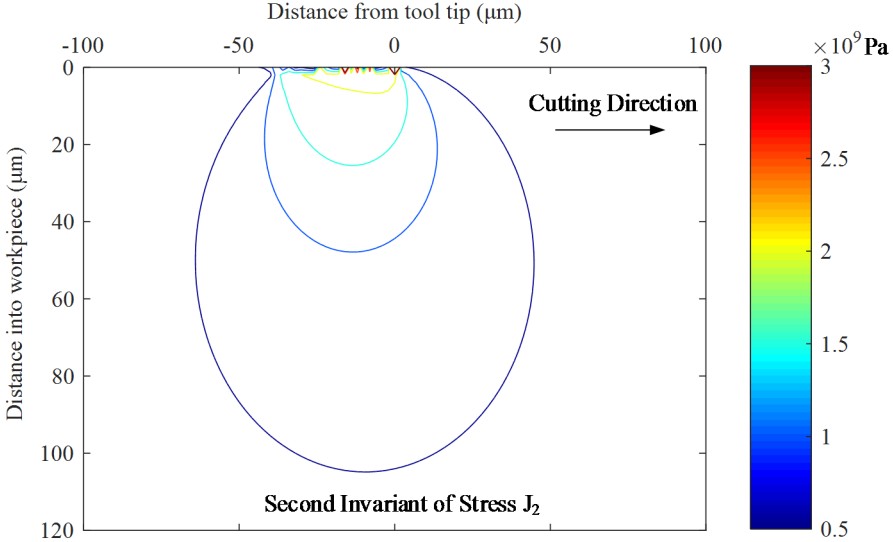

**Figure 7.** Stress field contours in workpiece.

### 4.3. Hybrid Algorithm for Stress Loading Process

The stress $\sigma_{total}$ in the workpiece is discretized into $M$ steps, and the loading calculation is carried out in an incremental manner.

$$\Delta\sigma_{xx} = \frac{\sigma_{xx}^{total}}{M} \quad \Delta\sigma_{zz} = \frac{\sigma_{zz}^{total}}{M} \quad \Delta\sigma_{xz} = \frac{\sigma_{xz}^{total}}{M} \tag{28}$$

At the end of each loading step, it is necessary to determine whether or not yielding occurs. The TANH constitutive model is used for the yield stress calculation and the subsequent yield surface is updated by combining with the kinematic hardening criterion. Its expression is:

$$VM = \frac{3}{2}\big(s_{ij} - \alpha_{ij}\big)\big(s_{ij} - \alpha_{ij}\big) - K^2 \tag{29}$$

where $s_{ij} = \sigma_{ij} - (\sigma_{kk}/3)\delta_{ij}$, where $s_{ij}$ is deviatoric stress, $\delta_{ij}$ is Kronecker delta. $\langle\,\rangle$ is the Macaulay bracket as, $\langle x \rangle = 0.5(x + |x|)$, $n_{ij}$ is the unit normal vector pointing outward the yield surface, which can be expressed as:

$$n_{ij} = \frac{s_{ij} - \alpha_{ij}}{\sqrt{2}k} \tag{30}$$

When $VM < 0$, the material belongs to the elastic loading state. When $VM = 0$, the material belongs to the plastic loading state. The elastic strain is expressed as:

$$\Delta\varepsilon_{ij}^{e} = \frac{1}{E}\Big[(1 + v)\Delta\sigma_{ij} - \delta_{ij}v\Delta\sigma_{kk}\Big] \tag{31}$$

where $E$ is the elastic modulus and $v$ is Poisson's ratio and $\Delta\sigma_{kk} = \Delta\sigma_{xx} + \Delta\sigma_{yy} + \Delta\sigma_{zz}$.

According to the plastic flow criterion, plastic strain is expressed as:

$$\Delta\varepsilon_{ij}^{p} = \frac{1}{h}\langle\Delta s_{kl}n_{kl}\rangle n_{ij} \tag{32}$$

where $h$ is a plastic modulus and represents a hardening rate of a material.

Since the stress of workpiece may have a large fluctuation range during the cutting process, the Hybrid function is introduced and expressed as:

$$\psi = 1 - \exp\left(-\kappa\frac{3h}{2G}\right) \tag{33}$$

where $\kappa$ is a constant, $G$ is the shear modulus of elasticity. $\Delta\sigma_{xx}$ and $\Delta\sigma_{yy}$ are stress increments in the cutting direction and perpendicular to cutting direction respectively, which can be expressed as:

$$\begin{cases} \Delta\varepsilon_{xx} = \frac{1}{E}\left[\Delta\sigma_{xx} - v\left(\Delta\sigma_{yy} + \Delta\sigma_{zz}^*\right)\right] + \alpha_T\Delta T + \frac{1}{h}\left(\Delta\sigma_{xx}n_{xx} + \Delta\sigma_{yy}n_{yy} + \Delta\sigma_{zz}^*n_{zz} + 2\Delta\tau_{xz}^*n_{xz}\right)n_{xx} \\ \quad = \psi\left(\frac{1}{E}\left[\Delta\sigma_{xx}^* - v\left(\Delta\sigma_{yy} + \Delta\sigma_{zz}^*\right)\right] + \alpha_T\Delta T + \frac{1}{h}\left(\Delta\sigma_{xx}^*n_{xx} + \Delta\sigma_{yy}n_{yy} + \Delta\sigma_{zz}^*n_{zz} + 2\Delta\tau_{xz}^*n_{xz}\right)n_{xx}\right) \\ \Delta\varepsilon_{yy} = \frac{1}{E}\left[\Delta\sigma_{yy} - v(\Delta\sigma_{xx} + \Delta\sigma_{zz}^*)\right] + \alpha_T\Delta T + \frac{1}{h}\left(\Delta\sigma_{xx}n_{xx} + \Delta\sigma_{yy}n_{yy} + \Delta\sigma_{zz}^*n_{zz} + 2\Delta\tau_{xz}^*n_{xz}\right)n_{yy} = 0 \end{cases} \tag{34}$$

where $\Delta\sigma_{xx}^*$, $\Delta\sigma_{yy}^*$, $\Delta\sigma_{zz}^*$ and $\Delta\tau_{xz}^*$ are the stresses obtained by Equation (27). Integrating the history of stress increment can obtain the stress distribution inside the workpiece after the stress loading process.

*4.4. Stress Release Process*

After the stress loading process is completed, the stress release process is performed to satisfy the boundary conditions. And the residual stress is the stress remaining inside the workpiece. According to the model proposed by Merwin and Johnson, the internal stress and strain of the workpiece after stress release process need to satisfy the following boundary conditions:

$$\begin{array}{cccc} \varepsilon_x^r = 0 & \varepsilon_y^r = 0 & \varepsilon_z^r = f(z) & \gamma_{xz}^r = f(z) \\ \sigma_x^r = f(z) & \sigma_y^r = f(z) & \sigma_z^r = 0 & \tau_{xz}^r = 0 \end{array} \tag{35}$$

The release increment for each step is expressed as:

$$\Delta\sigma_z = -\frac{\sigma_z^r}{M} \quad \Delta\tau_{xz} = -\frac{\tau_{xz}^r}{M} \quad \Delta\varepsilon_{xx} = -\frac{\varepsilon_{xx}^r}{M} \quad \Delta T = -\frac{T^r}{M} \tag{36}$$

As with the stress loading process, it is also necessary to determine whether the workpiece has yielded during the stress release process. When $VM < 0$ or $VM = 0$ and $ds_{ij}n_{ij} \geq 0$, calculate the stress increment under elastic release by Equation (37):

$$\begin{cases} \Delta\sigma_x^r = \frac{E\Delta\varepsilon_x + (1+v)(\Delta\sigma_z v - E\alpha_T\Delta T)}{(1-v^2)} \\ \Delta\sigma_y^r = \frac{vE\Delta\varepsilon_x + (1+v)(\Delta\sigma_z v - E\alpha_T\Delta T)}{(1-v^2)} \end{cases} \tag{37}$$

When $VM = 0$ and $ds_{ij}n_{ij} < 0$, calculate the stress increment under plastic release by Equation (38):

$$\begin{cases} \Delta\sigma_y^r = \frac{\left(-\frac{v}{E} + \frac{1}{h}n_xn_y\right)(C - \alpha_T\Delta T) - \left(\frac{1}{E} + \frac{1}{h}n_xn_x\right)(D - \alpha_T\Delta T)}{\left(-\frac{v}{E} + \frac{1}{h}n_xn_y\right)^2 - \left(\frac{1}{E} + \frac{1}{h}n_xn_x\right)\left(\frac{1}{E} + \frac{1}{h}n_yn_y\right)} \\ \Delta\sigma_x^r = \frac{D - \left(\frac{1}{E} + \frac{1}{h}n_yn_y\right)\Delta\sigma_y^r - \alpha_T\Delta T}{-\frac{v}{E} + \frac{1}{h}n_xn_y} \end{cases} \tag{38}$$

where

$$\begin{cases} C = \Delta\varepsilon_x^r + \left(\frac{v}{E} - \frac{1}{h}n_xn_z\right)\Delta\sigma_z^* - \frac{2}{h}\Delta\tau_{xz}^*n_{xz}n_x \\ D = \left(\frac{v}{E} - \frac{1}{h}n_yn_z\right)\Delta\sigma_z^* - \frac{2}{h}\Delta\tau_{xz}^*n_{xz}n_y \end{cases} \tag{39}$$

The residual stress obtained in the previous step is used as the initial stress for the next step until the stress and strain enter a stable state. The calculation result is the residual stress inside the workpiece after the cutting process is completed. The flow chart of calculating the residual stress is shown in Figure 8.

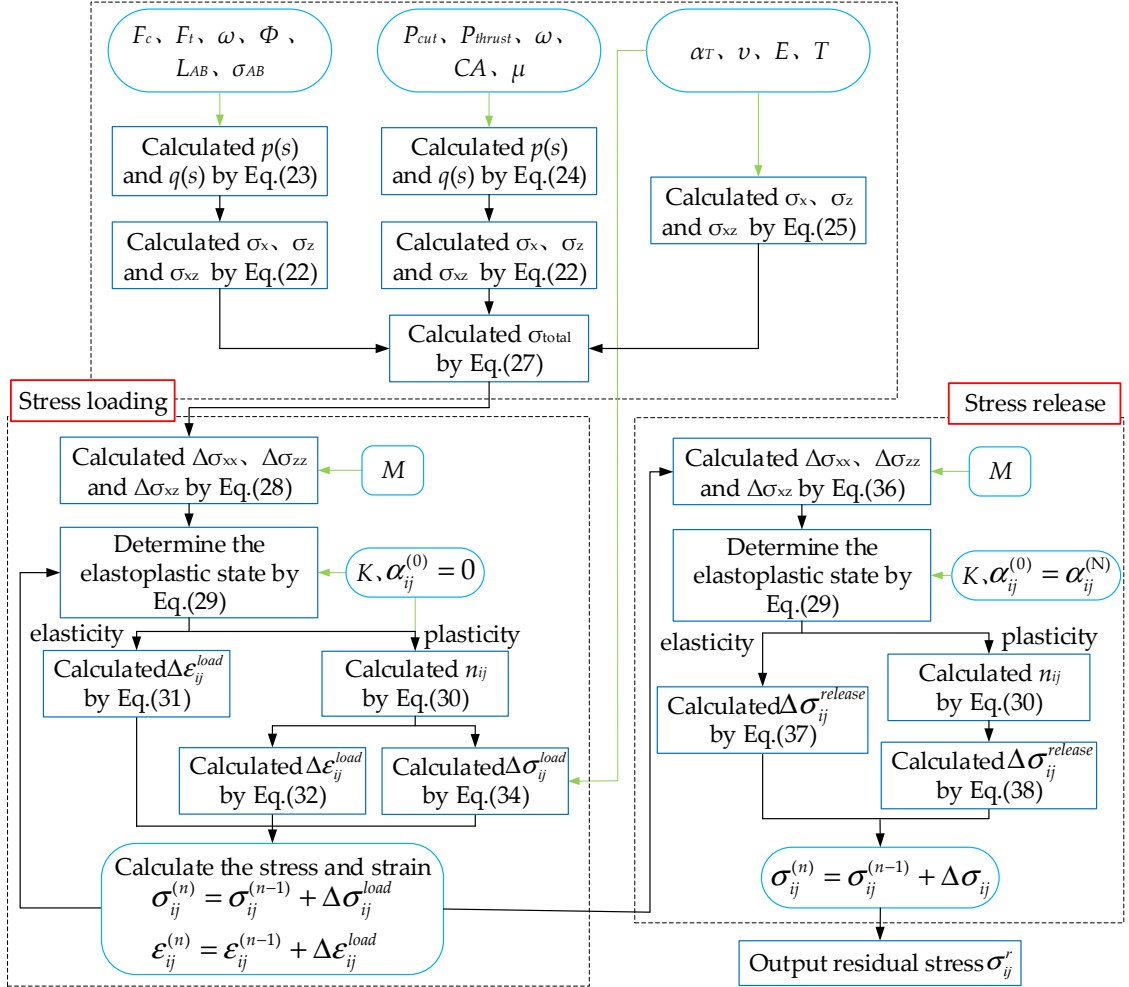

**Figure 8.** Flow chart for calculating residual stress.

## 5. Experimental Validation of Milling Force Model and Residual Stress Model

### 5.1. Experimental Machine Tool, Tool and Workpiece Material

The experiments are carried out on a VDL-1000E three-axis CNC machining center, which is made in Profile of General Technology Group Dalian Machine Tool Corporation, Dalian, China. The range of the spindle speed is 45–8000 rpm, and the feed rates in the three directions of X, Y, and Z are 24/24/18 m/min. The tool is an uncoated 2-blade carbide end tool (ANSI: 2P160-1000-NA H10F) produced by Sandvik. The diameter of the tool is 10mm, the helix angle $\beta$ is 25°, the rake angle $\alpha$ is 13.5°, the tool corner radius is 8 μm, the total length is 90 mm, and the length of cutting part is 32 mm. The workpiece to be machined is a rectangular block of Ti6Al4V alloy with a size of 50 mm × 50 mm × 10 mm, and the accuracy of surface polish is Ra 0.4 μm.

The experimental site is shown in Figure 9. The force signal during milling is measured by kistler5236b (Kistler Group, Winterthur, The Switzerland), in which force sensor is composed of rotor and stator. The rotor is the tool handle, which is used to clamp the tool. The stator is the signal receiving end, which receives the signal measured by the rotor. The uncertainty of the measurement is in the range of 1% of the measured value. The workpiece is fixed on the machine tool guideway with a fixture, and the machining method is down milling without cutting fluid. To avoid cutting vibrations and obtain high surface quality, the experimental scheme is shown in Table 2.

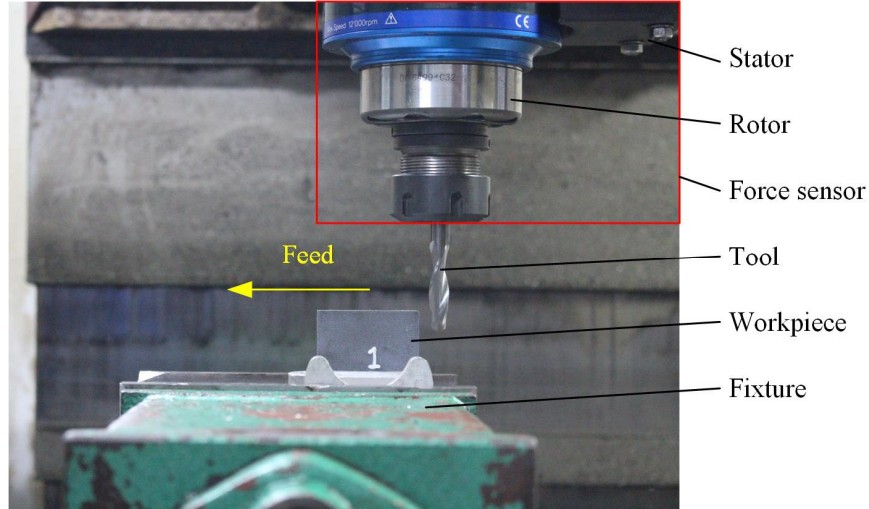

**Figure 9.** The experimental site.

The material parameters needed to be input into the prediction model are divided into two parts: tool material parameters and workpiece material parameters. The material parameters of workpiece (Ti6Al4V) and tool (WC) refer to the research of the literature [14], as shown in Table 3.

**Table 3.** Material parameters of workpiece and tool [14].

| Material Parameter | Ti6Al4V | WC |
|---|---|---|
| $\rho$ (kg/m3) | 4430 | 15630 |
| K (W/m·°C) | $7.039e^{0.0011T}$ | 55 |
| $C_p$ (J/kg °C) | $505.64e^{0.0007T}$ | $0.32T + 1.32 \times 10^{-4}$ |
| $T_{melt}$ (°C) | 1650 | - |

### 5.2. Experimental Validation of Milling Force Model

The comparison results between the experimental data and the predicted values of the milling force under the four sets of experimental parameters shown in Table 2 are shown in Figure 10. By the average value $\varepsilon_r^*$ of the absolute value of the error between the experimental data and the predicted value, the prediction accuracy of the model is quantitatively evaluated. $\varepsilon_r^*$ is calculated by the following equation:

$$\varepsilon_r^* = \frac{1}{n} \sum_{i=1}^{n} \left| F_m^i - F_e^i \right| \tag{40}$$

where $F_m^i$ is the predicted value of milling force, $F_e^i$ is the experimental value of milling force, $n$ is the number of measured experimental data. The prediction error $\varepsilon_r^*$ is shown in Table 4. The cutting force prediction value in X direction is more accurate.

**Table 4.** Prediction error $\varepsilon_r^*$ in cases 1–4.

| Case | $\varepsilon_r^*$ in X Direction (N) | $\varepsilon_r^*$ in Y Direction (N) | $\varepsilon_r^*$ in Z Direction (N) |
|---|---|---|---|
| 1 | 10.61 | 37.64 | 2.53 |
| 2 | 22.06 | 51.49 | 4.57 |
| 3 | 15.33 | 52.75 | 2.60 |
| 4 | 13.39 | 37.25 | 3.06 |

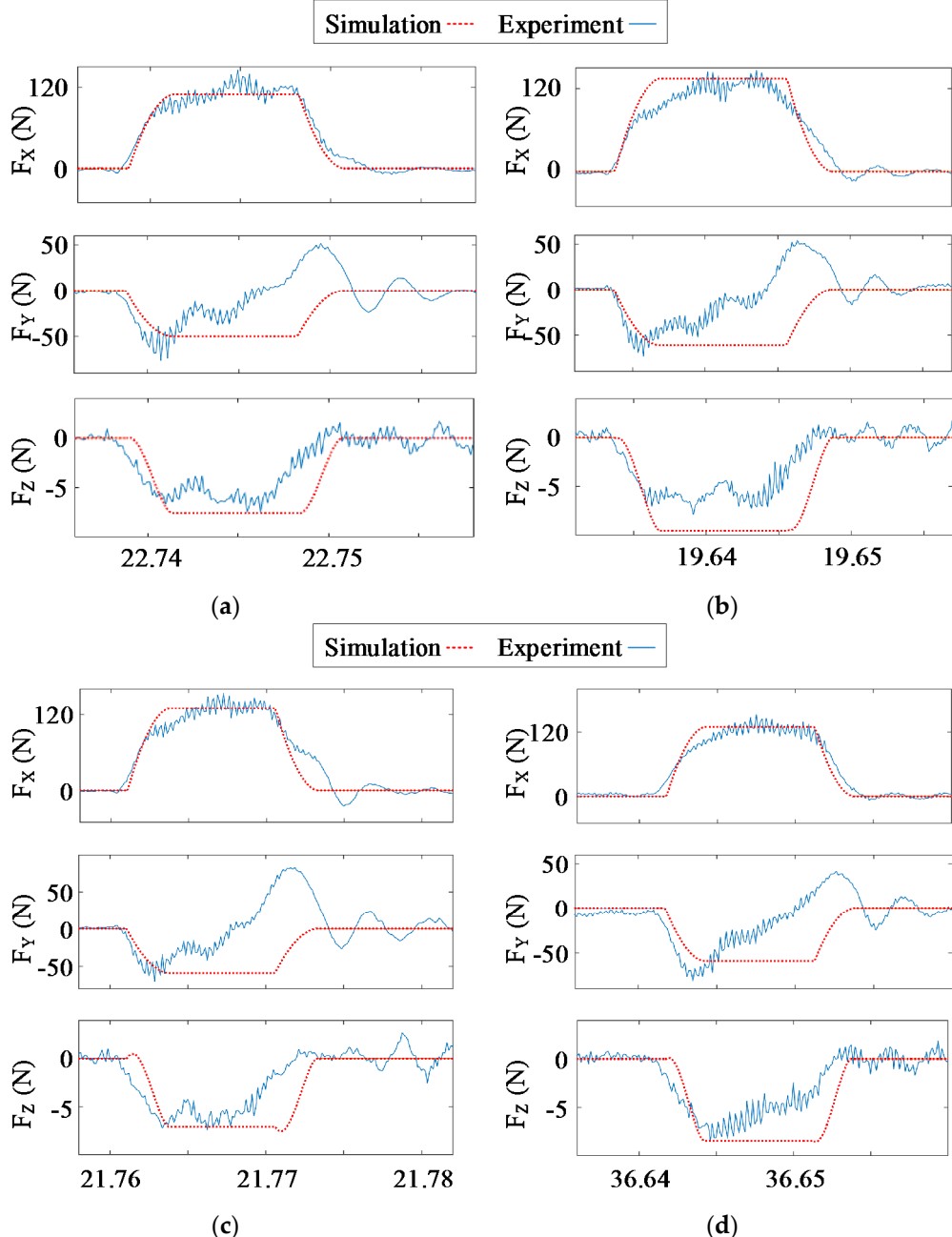

**Figure 10.** Comparison of predicted value and experiment value of milling force $F_X$, $F_Y$ and $F_Z$ in cases 1–4. (**a**) Case 1; (**b**) Case 2; (**c**) Case 3; (**d**) Case 4.

According to Figure 10, it can be found that the prediction error of the milling force $F_Y$ is large, and $\varepsilon_r^*$ can reach 52.75 N. When entering the stable cutting stage, $F_Y$ suddenly presents a fluctuating state. This is due to the poor rigidity of the tool (or workpiece), resulting in the cutter-drawing, which further caused the vibration between the tool and the workpiece. Compared with the experimental data of milling forces in other two directions, it can be found that there is still a wave value of convergence in the milling force $F_Y$ after the tool cuts out from the workpiece, which proves that there is cutting vibration in the direction perpendicular to the workpiece.

The increase of cutting speed increases the milling force $F_X$ and $F_Y$, but it has little effect on the milling force $F_Z$. Therefore, the experimental value of the cutting force $F_Z$ in case 2 is less than the predicted value. The increase of cutting width causes greater cutting vibration, and the peak value of milling force $F_Y$ in case 3 increases significantly, which shows that the cutting vibration is enhanced.

The milling force model divides the milling process into three cutting stages, and the experimental data show that the milling force fluctuates obviously in the stable cutting stage. Figure 11 illustrates the contact relationship between the cutting edge and the workpiece in the stable cutting stage. $t_1$ and $t_2$ are different time points in the same stable cutting stage, corresponding to two different cutting states. Considering the cutter-drawing due to excessive tool overhang during milling, it is considered that in the stable cutting stage, the contact area between the cutting edge and the workpiece at time $t_2$ is higher in the axial direction than that at time $t_1$. The position of the milling force acting on the tool changes, which changes the degree of deflection. As a result, the milling force fluctuates during the stable cutting stage. In addition, it may also be due to friction chatter caused by friction between the tool and the workpiece in the same direction as the cutting speed [15].

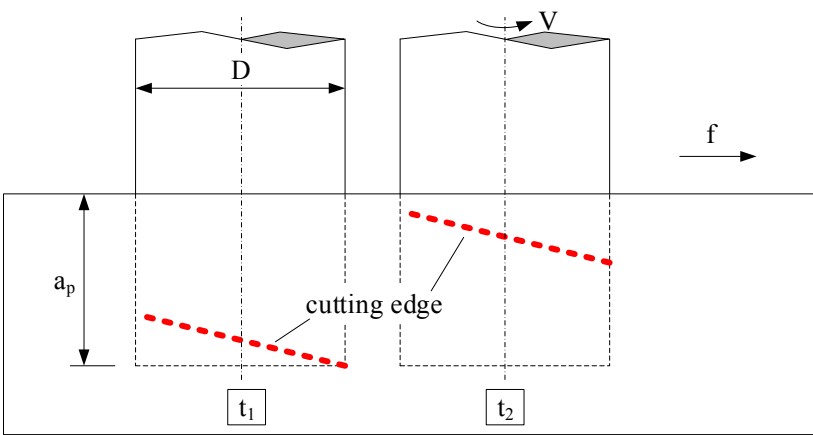

**Figure 11.** Schematic diagram of the contact relationship between the tool and the workpiece in the stable cutting stage.

### 5.3. Measurement Device and Measurement Scheme for Residual Stress

The instrument for measuring the residual stress is an X 'Pert-Pro X-ray diffractometer and the test is performed using method $\sin^2 \psi$, as shown in Figure 12. For the residual stress test of Ti6Al4V alloy, Cu target was selected as the X-ray tube. The XRD stress test parameters for Ti6Al4V are given in Table 5.

**Table 5.** The XRD stress test parameters for Ti6Al4V.

| Plane (hkl) | Radiation | Bragg's Angle $2\theta$ | 1/2 $S_2$ (MPa$^{-1}$) | $S_1$ (MPa$^{-1}$) | Voltage (kV) | Current (mA) |
|---|---|---|---|---|---|---|
| (213) | Cu-K$\alpha$ | 141.3° | $12.18 \times 10^{-6}$ | $-3.09 \times 10^{-6}$ | 40 | 40 |

Based on the theory of residual stress prediction algorithm, it is considered that under ideal conditions, the residual stress distribution at any point on the surface of the workpiece is consistent. In order to facilitate the measurement of the residual stress and the electrolytic etching in the machined surface, the position with a depth of 4 mm along the blade axis was selected for measurement. According to the milling force data shown in Figure 10, the stage where the milling force is relatively stable is selected for measurement. When electrolytic etching the surface, the selected electrolyte is saturated NaCl solution, the voltage is set to 15 V, the diameter of the nozzle is 15 mm, and the corrosion depth is measured by a SKCH-1A thickness gauge after each electrolytic corrosion.

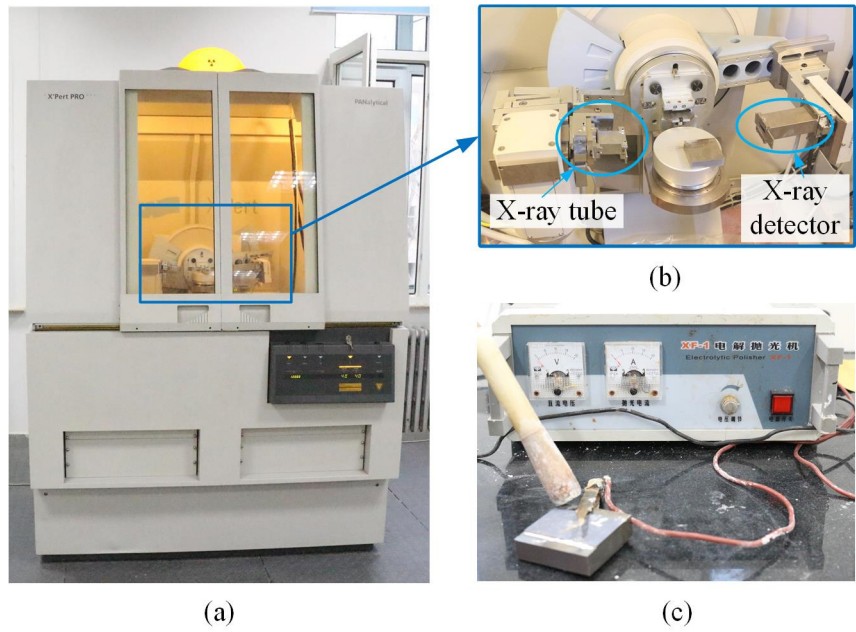

**Figure 12.** The instrument for measuring the residual stress. (**a**) X 'Pert-Pro X-ray diffractometer; (**b**) Table of X-ray diffractometer; (**c**) Electrolytic polisher.

### 5.4. Experimental Validation of Residual Stress Model

The predicted and experimental values of the residual stress under the experimental parameters of case 1 and case 3 are shown in Figure 13, it can be seen that the influence depth of residual stress is about 50 μm and the depth of the peak value of subsurface residual stress is between 15 μm and 20 μm. The variation of residual stress $\sigma_x$ and $\sigma_y$ with depth is basically the same. The predicted value converges faster than the experimental value, which may be caused by the simplified method proposed in Figure 6. Because the model only considers the stress distribution between AB two points, the response depth of the predicted residual stress is smaller than the experimental residual stress.

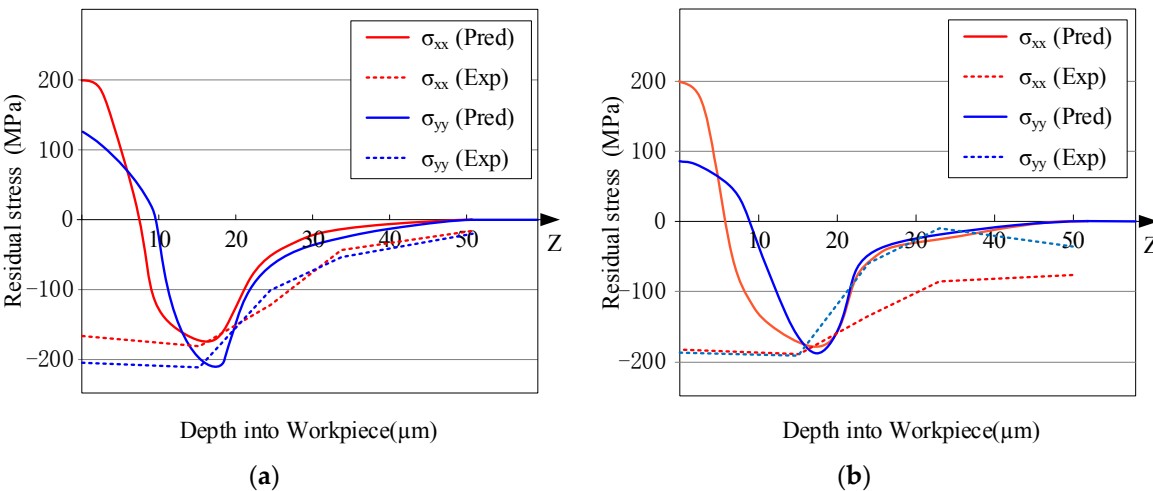

**Figure 13.** Prediction of residual stress distribution in the machined surface. (**a**) Case 1; (**b**) Case 2.

When the depth into workpiece is 0 μm, the predicted value of residual stress is different from the experimental value. The predicted value is tensile stress, but the experimental value is compressive stress. In the surface residual stress measurement of most machined workpieces, surface oxidation may be an important cause of prediction error. Moreover, the prediction model of residual stress in this paper does not consider the effect of phase transformation. For Ti-6Al-4V materials, the phase

transformation from α to β will lead to the increase of grain volume, which will lead to the greater compressive residual stress on the surface. Similarly, it also explains that the residual stress of the surface tends to be tensile stress.

The peak value of residual stress in the machined surface has the greatest effect on the fatigue life of the workpiece. It is particularly important to accurately predict this feature. Figure 14 shows the predicted and experimental values of the residual stress under the four sets of process parameters. The residual stress prediction algorithm is based on the plane stress assumption, so the stress $\sigma_z$ perpendicular to the surface of the workpiece is not considered. The numbers in Figure 14 indicate prediction errors. As shown in Figure 14, the residual stress prediction model can accurately predict the peak value of the residual stress in the subsurface. The predicted error of case 3 and case 4 are relatively large. Comparing the milling force $F_Y$ in Figure 10, it can be found that in case 3 and case 4, the cutting width and the feed rate are increased, which enhances the cutting vibration during the stable cutting stage, resulting in changes in the stress distribution in the machined surface. This may be one of the reasons for the prediction error of residual stress.

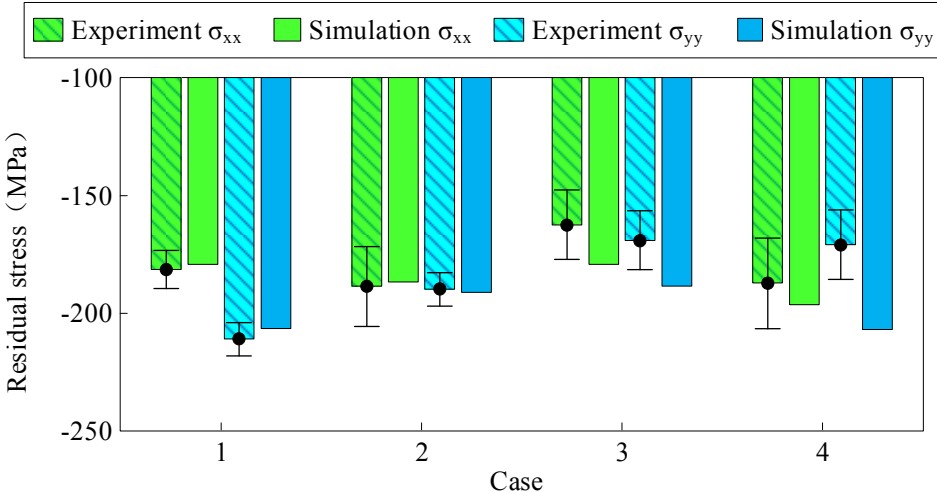

**Figure 14.** Comparison between the predicted and the experimental peak value of the residual stress in the subsurface.

## 6. Conclusions

In this paper, the milling force and residual stress model of titanium alloy are established with considering the coupling effect of thermal-mechanical load. In this model, the complex contact relationship between tool and workpiece and the time-varying cutting thickness model are considered. The conclusions are as follows:

(1) The contact relationship between the tool and the workpiece during the milling is analyzed in detail. Compared with the milling forces $F_X$ and $F_Z$, it can be found that there is still a wave value of convergence in the milling force $F_Y$ after the tool cuts out from the workpiece, which proves that there is cutting vibration in the direction perpendicular to the workpiece. It is also found that the increase of radial cutting depth and feed rate will significantly increase the tool vibration. In the direction perpendicular to the workpiece surface, the milling force in the stable cutting stage fluctuates due to the influence of the cutter-drawing, but the milling forces $F_X$ and $F_Z$ can still be predicted well.

(2) In order to use the thermal coupling iterative algorithm and realize the rapid prediction of the peak value of the residual stress, the geometry of the workpiece surface is simplified. The residual stress model accurately predicted the peak value of the residual stress in the subsurface of the workpiece. With the increase of the radial cutting depth and feed rate, the cutting vibration will

be intensified, the stress state in the surface of the workpiece will change, and the prediction error of the residual stress will increase.

(3)  According to the prediction results of residual stress, it can be seen that the influence depth of residual stress is about 50 μm and the depth of the peak value of subsurface residual stress is between 15 μm and 20 μm. The variation of residual stress *σx* and *σy* with depth is basically the same. The stress in the machined surface is tensile stress, and the peak value of subsurface stress is compressive stress, which may be due to the influence of cutting heat, resulting in a large tensile stress in the surface.

**Author Contributions:** Conceptualization: C.Y., X.H. and S.Y.L.; formal analysis: X.H. and C.Y.; investigation: C.Y. and X.H.; supervision: C.Y., X.L., S.Y.L., X.J. and L.W.; validation: C.Y., X.H., X.L. and F.Y.; writing-review and editing: X.H. and C.Y. All authors have read and agreed to the published version of the manuscript.

**Funding:** This research was funded by Projects of International Cooperation and Exchanges NSFC (Grant Number 51720105009), National Key Research and Development Project (Grant Number 2018YFB2002201), Natural Science Outstanding Youth Fund of Heilongjiang Province (Grant Number YQ2019E029) and Outstanding Youth Project of Science and Technology Talents (Grant Number LGYC2018JQ015).

**Conflicts of Interest:** The authors declare no conflict of interest.

## Nomenclature

| | |
|---|---|
| $a_c$ | chip thermal diffusivity |
| $B(x)$ | percentage of heat generated at the tool–chip interface entering the chip |
| $C_p$ | Specific Heat Capacity |
| $E$ | elastic modulus of the workpiece material |
| $F$ | friction force at the rake face of the tool |
| $f$ | feed rate |
| $f_z$ | feed per tooth |
| $G$ | shear modulus of elasticity |
| $h$ | plastic modulus |
| $h_c$ | cutting thickness |
| $K_c$ | Thermal conductivity of the chip |
| $K_t$ | Thermal conductivity of the tool |
| $L_{AB}$ | shear plane length |
| $L_c$ | contact length between tool and chip |
| $N$ | normal force at the rake face of the tool |
| $n_{ij}$ | unit normal vector pointing outward the yield surface |
| $P_{thrust}$ | plowing force normal to the newly generated surface |
| $P_{cut}$ | plowing force in the cutting direction |
| $P_{total}$ | resultant cutting force |
| $q_{shear}$ | heating source intensity in the primary shear zone |
| $q_f$ | heating source intensity in the second deformation zone |
| $q_{rub}$ | heating source intensity in the tertiary deformation zone |
| $R$ | tool radius |
| $s_{ij}$ | deviatoric stress |
| $T_0$ | initial temperature |
| $T_{melt}$ | melting point |
| $t$ | cutting time |
| $V_c$ | chip velocity |
| $v$ | Poisson's ratio of the workpiece material |
| $w$ | rotational speed of tool |
| $w_t$ | width of the cutting edge participating in cutting |
| $z$ | axial height of the tool |
| $\alpha$ | rake angle of the tool |

| $\alpha_T$ | thermal diffusion coefficient of the workpiece material |
|---|---|
| $\beta$ | tool helix angle |
| $\lambda$ | friction angle of the rake face. |
| $\Phi$ | shear angle of the primary shear zone |
| $\theta$ | angle between the force $P_{total}$ and the shear plane AB |
| $\sigma_{AB}$ | flow stress on the shear plane |
| $\gamma$ | percentage of heat generated by the heat source $q_{rub}$ entering the tool |
| $\rho$ | density |
| $\varphi_{int}$ | cut-in angle |

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
