# Peer review of "Analytical Prediction of Residual Stress in the Machined Surface during Milling"

_metals, doi:10.3390/met10040498_

Round 1

Reviewer 1 Report

Analytical Prediction of Residual Stresses in milling is and interesting subject that this manuscript deals with. However, there are some points to be clarified by the authors:

The authors have to include a table with the nomenclature employed.

What software has been employed to obtain the results?

The authors mention (line 125) that "considering the thermal-mechanical coupling, a milling force model is established". However, this is not clearly shown in Section 3. Thermo-mechanical coupling model is not presented in this Section. How can the heat generated in the milling operation be predicted? How is the chip formation supposed to be formed? Strain-rate and strain hardening? etc. On the other hand, Friction conditions? How are they modelled? etc. This has to be clearly shown and appropriate references have to be included.

In Section 4.1. The authors have to clearly shown the novelty of their study in relation to previously published studies. Equation (17) needs to be justified. The authors have to provide appropriate references for this equation as well as for the rest of the equations of this Section. The authors should include a graph showing the boundaries for the stress distribution.

Section 4.2 Equation (20) has to be justified. The authors have to provide appropriate references for this equation.

Section 5. Experimentation

How is the force measured? The employed equipment is not detailed. Precision of measurements?

The authors should either modify Figure 9 or add another figure in order to show the relative error (experimental vs predicted).

How were the data shown in Table 4 obtained?

Experimental results are not shown. These results should be plotted in a graph along with the error in the measurements. That is, a graph (residual stresses vs depth into workpiece) experimentally obtained should be included for both transverse and cutting directions and results should be analysed.

Figure 13 is not justified. What are the values of the measuring errors?

Reviewer 2 Report

Observe markings and symbols especially tool geometry and technological parameter. I recommend using ISO 3002 in figures and equations. The symols in the pictures do not correspond to the median.
Marking errors in figure 2 symbol t1 is 2x, alpha is not the rake angle in figure 1, and others.
Check English eg. “Third should be tertiary”, “residual stress distribution is consistent” what is meant by consistent?

What is meant by “simulation” in the text? It is a simulation eg. FEM, then I recommend adding the description and conditions of the simulation. In case of calculations according to equations in the paper, I recommend to complete the results in the form of eg. table.

Explain on what basis the cutting parameter was chosen? No plan experiment. And why was the experiment done without cutting fluid?
At the beginning of the article I would expect to make a scientific hypothesis, I recommend adding.

formally
Line 420 The line before Table 2 is missing.
Line 374 Missing the beginning of the sentence “is calculated by the”
Line 356 The angle character ° is on the new line, there should be no space between the number and °
Line 355 - tool specification - total length / length of cutting part missing

Overall assessment
The article has potential.
I propose to change Conclusions slightly. In my opinion, there are no proper details in the Conclusions. Conclusions are somehow simplistic as it seems to be observational without revealing findings of generic academic value. What I mean that based on the results of some generic and fundamental academic conclusions need to be drawn. In Conclusions, please try to emphasize the novelty, put some quantifications and comment on the limitations. The conclusions should also highlight the progress in understanding the knowledge presented in the work.

Reviewer 3 Report

Review:  Analytical Prediction of Residual Stress in the Machined Surface During Milling

Caixu Yue,·Xiaole Hao,·Xia Ji,·Xianli Liu,·Steven Y. Liang,·Lihui Wang·and Fugang Yan

Comments to author:

Short summary:

The publication deals with the prediction of residual stresses in milled Ti6Al4V components by an analytical model. An existing model for the determination of the cutting force according to Oxley and Stevenson as well as an algorithm for thermo-mechanical coupling during turning is used. For milling applications, these models are adapted to the different cutting conditions. With this adapted model, the forces during milling can finally be calculated. Then the consideration of mechanical and thermal influences in the stress distribution in the component is shown and the analytical calculation of residual stresses is presented in detail. Finally, experimental investigations are carried out on Ti6Al4V, residual stress depth profiles are measured by X-ray diffraction and compared with the results of the simulation.

Abstract

The abstract describes the topic well, but could mention the models used by other authors and refer to the expected results.

Keywords

Keywords are adequate

Linguistic expressions

It is generally recognized that the mother tongue of the author is not English. This is particularly conspicuous by the fact that in some places the choice of words is not precise. Furthermore, the same words and phrases are often used in many places, which after some time greatly disturbs the reading flow.

It is clearly pointed out that a native English speaker should proofread the document

Tables and Figures

In general, the illustrations are of uniform quality and their presentation is appropriate to the content.  However, there are the following remarks:

Fig. 1 top left: missing labelling of the Y-axis,

Fig. 5: All references to equations in the text are incorrect.

Fig. 6: Some names begin with lower case letters and others with upper case letters. Ensure a uniform procedure in all other figures.

Fig. 7: All references to equations in the text are incorrect. Many cannot be found in the text under other references

Fig 9: An enlargement of the section of the force measurement record would help to better evaluate the results.

Fig 10: In Figure 10 it is not clear what is meant by the times t1 and t2. The text also does not describe this clearly.

Scientific Quality

The publication deals with a very interesting topic that links different existing models and transfers them to milling. The introduction is very extensive and the publications by Ji Xia and Oxley are discussed. A flowchart like the one in the middle of the publication would be helpful at this point to show the reader the procedure and use of the various models to facilitate understanding. The exact procedure is very difficult to understand and to follow.

The innovations of the publication are explicitly mentioned at the end of the introduction. However, the second innovation is difficult to find in the manuscript and should either be revised or more addressed in the text.

The presentation of the used equations is very extensive in the following and is also done at the beginning by mentioning the original authors. The adaptation of the existing models is also carried out with the help of visual representation and is comprehensible. From Chapter 4, however, it is very difficult to decide where the given equations come from. Here a reference to the original source is necessary to evaluate the procedure. Otherwise a detailed derivation is missing.  

The validation of the simulation results by milling Ti6Al4V is a very good method. And also the measuring methods are basically suitable. However, it is essential to determine the grain size of Ti6Al4V in order to evaluate whether an X-ray residual stress measurement is possible at all.

Another unpleasant fact is that the radiographic measurement results are not also published and only the deviation of the range of maximum residual compressive stresses is compared. A direct comparison of the measured values with the simulated results would help the publication even more.

Irrespective of the above-mentioned points, the following inconsistencies have been noted:

  1. 153 What does PTotal stand for?
  2. 325 Is the 4 times listing of Δs*xx correct?

Adequacy of the reference

References are adequate from renowned Journals.

Result of Review:

Major revision is recommended.

Reviewer 4 Report

The paper presents an original analytical model to predict residual stresses induced in milling.

The proposed scientific methodology is of high interest. However several bug improvements have to be made before being acceptable for publication.

The paper starts with a presentation of a list of experimental works. The coherence and the added value of this section is highly questionable. Some papers are related to turning, others to milling or ball milling. the workmaterial differs from a paper to another. Moreover they do not reports comprehensively the "major" papers in this field. As this paper deals with side milling, only experimental papers dealing with side milling should be reported and analysed.

Regarding the state of the art in residual stress modelling, authors should read carefully the PROCEDIA CIRP published after the last 5 editions of the CIRP conference on Surface Integrity, as well as the CIRP Annals published during the last 10 years. They would be able to make an effective state of the art about the 'edge' research groups working on analytical and numerical approaches in residual stresses.

regarding the state of the art in analytical modeling, there many self-citation. it is questionable why references 12, 13 and 17 are reported. They have no link with the topic of the present paper. On the contrary the analysis of references 14, 15, 16 and 18 have to be detailed so as to justify the originality of the present article.

Moreover they report papers dealing with the influence of tool wear and vibration. But, in their paper, they do not consider tool wear and vibration. What is the interest to report these references from the litterature ?

In the introduction, there is no figure presenting the milling operation that will be investigated. As they are many milling operations (side milling, face milling, trochoidal milling, peck milling, ...), it is mandatory to make a clear drawing and to introduce the geometrical and kinematic parameters considered in the paper. Plot also the force components monitored.

Which grade of "titanium" is investigated ?

What the material of the cutting tool ?

Do they consider a cutting fluid ? its cooling effect ?

Figure 1 shows that the influence of the cutting edge preparation is not considered in their model. The influence of the rubbing phenomena at the interface between the flank face and the workmaterial are not considered. These two assumptions have a huge influence as they avoid the two main region responsible for residual stress generation. The phenomena in the primary shear zone are only of secondary importance. As a consequence, it is strongly recommended to make a justification of these assumptions.

Teh analytical model consider a friction coefficient. What is its value ? Have you checked the state of the the most advanced research work in modelling and characterization of friction in metal cutting ? especially when machining titanium ?

Section 3 : add a figure explaining how cutting edges are discretized and how elementary straight oblique cutting edges are extracted.

Figure 4 : where is 'hc' on the figure ?

In section 4, the effect of the rubbing on the flank face is back in the model. Why not in Figure 1. How friction on this face has been considered ? calibrated ? It is a crucial parameter that controls the heat generation and by the way the tensile stresses on the surface. Moreover, what is the heat partition at the interface ?

Section 5. Which rake angle is considered ? Please use the terminology of the ISO standard as there is a huge difference between the normal and the orthogonal section.

surface roughness "Ra0.4m". Are you sure ?

Section 5 reports "down milling", whereas section 4 indicates "climb milling"; Where is the reality ?

problems with lines 373 and 374

Figure 9 are much too small to make a qualitative comparison of forces. Make it bigger or add a local zoom.

line 388 : the force is "52.745N". Are you able to measure a force with an accuracy of 0.001 N ? I do not believe ! Please provide reasonnable experimental values in agreement with the accuracy of your measuring device. What about the deviation of the measurements ? how many replication ?

Regarding the residual stress measurements, what is the dimension of the Xray beam ? what is his penetration depth ?
Have you averaged the results obtained by the analytical model so as to make a realistic comparison ?

How many Xray measurements have been done ? what is the deviation ?

What is the X and Y direction ? feed direction ? axial direction ? make a drawing

The basement of scientific paper is the capacity of a reader to make the calculation by him self to check that the results are correct. Fore sure, there is a lack of input data in the paper (flow stress model of titanium, ...).

Beside these criticisms, I really think that it is a promising work and that the paper has to be improved so as to fit the requirements of a future reference paper.

Round 2

Reviewer 1 Report

- The authors should indicate the version of MatlabTM they are using.
- The precision of measurements is not detailed. What is the uncertainty of the measurements?
- A graph showing the boundaries for the stress distribution has not been included in Section 4.1. This has to be done.
- Write units using () instead of /. For example, in Figure 9, write Fx (N) instead of Fx/N and so on. The same in Table 4, etc.
- When writing units, authors should place a space between the mangnitude and the corresponing unit. For example, in line 376, write 52.75 N instead "52.75N". This has to be modifed throughout the manuscript for all the magnitudes and units.

Reviewer 3 Report

L 346 Check the correct notation of the force-measuring unit.

L 350 How can the experimental scheme prevent vibrations?

L 357 Unpleasant page break

L 372 Unpleasant page break

Fig 10 Missing center Line in the left tool
